# Thermal Oviposition Performance of the Ladybird *Stethorus gilvifrons* Preying on Two-Spotted Spider Mites

**DOI:** 10.3390/insects14020199

**Published:** 2023-02-16

**Authors:** Maryam Jafari, Hossein Ranjbar Aghdam, Abbas Ali Zamani, Shila Goldasteh, Ebrahim Soleyman-Nejadian, Peter Schausberger

**Affiliations:** 1Department of Entomology, College of Agriculture, Arak Branch, Islamic Azad University, Arak 6134937333, Iran; 2Iranian Research Institute of Plant Protection, Agricultural Research, Education and Extension Organization, Tehran 1475744741, Iran; 3Department of Plant Protection, College of Agriculture, Razi University, Kermanshah 6718773654, Iran; 4Department of Behavioral and Cognitive Biology, University of Vienna, 1030 Vienna, Austria

**Keywords:** *Stethorus gilvifrons*, fecundity, *Tetranychus urticae*, temperature, non-linear modeling

## Abstract

**Simple Summary:**

This study models the effects of temperature on age-dependent fecundity of the ladybird *Stethorus gilvifrons*, which is an important predator of herbivorous spider mites such as *Tetranychus urticae*. Based on the fecundity of *S. gilvifrons* fed on *T. urticae*, examined at six constant temperatures from 15 to 34 °C, we generated thermal performance curves by using four non-linear models (Enkegaard, Analytis, Bieri-1, and Bieri-2). The thermal performance models presented here can be used to determine the optimal temperature for mass-rearing, simulate daily egg production, and predict field occurrence patterns and seasonal population dynamics of this predator. The models constitute an essential step towards developing temperature-based simulation models of the population dynamics of *S. gilvifrons* interacting with its spider mite prey.

**Abstract:**

The ladybird, *Stethorus gilvifrons* (Mulsant) (Coleoptera: Coccinellidae), is an important predator of two-spotted spider mites, *Tetranychus urticae* Koch (Acari: Tetranychidae), in southeastern Europe and western and southwestern Asia, such as Iran, India, and Turkey. To enhance forecasting the occurrence and performance of this predator in natural control and improve its usage in biological control, we evaluated and compared four non-linear oviposition models, i.e., Enkegaard, Analytis, Bieri-1, and Bieri-2. The models were validated by using data of age-specific fecundity of female *S. gilvifrons* at six constant temperatures (15, 20, 25, 27, 30, and 34 °C). All four models provided good fit quality to age-dependent oviposition at 15 to 30 °C (*R*^2^ 0.67 to 0.94; *R*^2^*_adj_* 0.63 to 0.94) but had a poor fit at 34 °C (*R*^2^ 0.33 to 0.40; *R*^2^*_adj_* 0.17 to 0.34). Within temperatures, the best performing models were Bieri-1 *(R*^2^), Bieri-2 (*R*^2^*_adj_*), and Analytis (RSS) at 15 °C, Bieri-1 at 27 °C, and Analytis at 20, 25, and 30 °C. Analytis was the best suited model across the wide temperature range tested (from 15 to 30 °C). The models presented here allow for prediction of the population dynamics of *S. gilvifrons* in field and greenhouse crops in temperate and subtropical climates.

## 1. Introduction

The two-spotted spider mite *Tetranychus urticae* Koch (Acari: Tetranychidae), which is a serious pest of numerous field and greenhouse crops, is notorious for its ability to develop resistance against synthetic chemical pesticides [1,2,3]. Attempts to control *T. urticae* using acaricides often fail, and thus, biological control using predatory mites of the family Phytoseiidae [4,5], and predatory beetles of the family Coccinellidae [6,7] has been promoted. Within the Coccinellidae, several species of the genus *Stethorus* have been investigated for their potential in the control of spider mites [8,9], such as *S. punctillum* Weise [10,11,12], *S. punctum* (Le Conte) [13], *S. madecassus* Chazeau [14], *S. pauperculus* Weise [15], and *S. gilvifrons* (Mulsant) [16,17]. The latter species has been reported from several regions in Europe, Africa, and Asia [18], and was reported for the first time in Iran in 1961 [19]. The general biological characteristics of *S. gilvifrons* such as its functional response, life table characteristics, and feeding activity have been studied [20,21,22], but little information is available on the effects of temperature on *S. gilvifrons* [23,24,25] and no temperature-dependent oviposition models have been developed. Compared with other *Stethorus* spp., *S. gilvifrons* has a very good searching activity, a short developmental time, relatively high survivorship, moderate longevity, high fecundity, and, in consequence, a high intrinsic rate of natural increase (*r_m_*). Therefore, it holds promising potential for mass-rearing and periodic augmentative releases in open field and greenhouse crops [25,26,27,28].

Environmental temperature is a critical abiotic factor that fundamentally affects the biology and ecology of ectothermic organisms such as insects and mites, regulating their behavior, physiology, and life history [29,30,31]. Accordingly, in biological pest management, the estimation of the developmental rate, longevity, survivorship, fecundity, and thermal thresholds of both herbivorous pests and their natural enemies has received great attention. Such efforts are important not only to further our understanding of their population dynamics but also for the development and optimization of integrated and biological pest management programs [32]. One important aspect when assessing and modeling temperature effects on the life history traits of ectothermic organisms is the choice of temperature regimes. Targeting the same mean temperatures, regimes can be constant (or close to constant with low amplitude fluctuations) or abrupt stepwise or smooth gradual with rare or frequent changes between temperatures. Such choices should be based on accuracy–generalizability–comparability trade-offs. Regimes that closely mimic natural fluctuations, for example [33,34], are a favorable choice if the objective is to accurately predict phenology in a given locality or region; close to constant regimes are easier to standardize and thus favorable for comparing the thermal requirements of populations of different geographic and climatic origins and/or species and/or data from different studies (for review [35]).

In previous studies [25,36], we assessed the effect of temperature on the life history of *S. gilvifrons* [25] and simulated the thermal developmental trajectories using linear and non-linear models [36]. Here we aimed at describing the thermal oviposition performance of *S. gilvifrons* by using four non-linear models [37,38,39,40,41] and comparing their fit to experimentally observed fecundity data at six constant temperatures [25]. Such oviposition models can be used to simulate the daily, age-dependent egg production of this predator under variable temperature conditions, which can be utilized in predicting egg occurrence patterns, in developing *S. gilvifrons* population dynamics models, and in establishing effective biological control strategies for spider mites. *Stethorus gilvifrons* is a promising biocontrol agent of spider mites in both greenhouse and field crops because it is able to develop and reproduce at relatively low (15 °C) and high (34 °C) temperatures [25] and may thus be suitable for use during early, peak, and late seasons.

## 2. Materials and Methods

### 2.1. Rearing and Experimental Conditions

The specimens of *S. gilvifrons,* which were used to found a laboratory population, were originally collected from two sugarcane fields (48°26′17.25″ E, 31°15′10.88″ N and 48°30′35.89″ E, 31°04′10.74″ N) (Amir-Kabir Agro-Industry Co.) located in Ahwaz region, Khuzestan province, Southwest of Iran. Individuals of the F2 generation of the field-sampled ladybirds were used in the experiment [25,33].

To establish a cohort, 20 *S. gilvifrons* individuals (males and females) were incubated at 27 °C on maize leaf discs (8 cm diameter) harboring mixed stages of *T. urticae*, resting on a moist cotton pad inside a transparent plastic container (19 × 14 × 4 cm). On the next day, newly laid eggs (<24 h old) of *S. gilvifrons* were collected and placed individually on maize leaf squares (4 × 4 cm) resting on moist cotton pads in plastic dishes (6 cm diameter). Both the plastic containers and plastic dishes were stored in growth chambers programmed at 15, 20, 25, 27, 30, and 34 °C (all ± 1 °C), 50 ± 5% R.H and a photoperiod of 16:8 h (L:D). The numbers per individual cohort (age < 24 h) at the starting point of the experiments were 200, 140, 120, 120, 120, and 120 eggs at the above temperatures [25,36]. Progress in development (hatching, molting, pupation, adult emergence) and survival were assessed every 24 h [23,36]. Due to juvenile survival varying with temperature [25], the number of adult females emerging at the above temperatures was 10, 20, 29, 33, 33, and 38 individuals. All emerged females were placed individually, together with a male, in a plastic container (19 × 14 × 4 cm) on new leaf squares containing mixed stages of spider mites as prey. During reproduction, the females were transferred daily to fresh leaf discs. The number of eggs laid was recorded daily until the female beetles died and their longevity (from adult emergence to natural death) was determined [25]. In case of male mortality, the dead male individual was removed and replaced by another similarly-aged male, taken from the laboratory rearing at the same temperature. Survival and fecundity of the females were recorded daily. Age-specific fecundity (*m_x_*) was calculated and used for modeling.

### 2.2. Non-Linear Models

Several non-linear models have been developed to describe the relationship between temperature and the developmental rate and/or fecundity of insects and mites [37,38,39,40,41,42,43,44,45]. Some of these models can be used for both developmental data and fecundity [41]. Among these, we selected the following four non-linear models to describe the relationship between temperature and age-dependent fecundity of *S. gilvifrons*.

### 2.3. Enkegaard Model

This non-linear model was suggested by [38] and used to fit fecundity data [38,40,41,45]. The equation of the Enkegaard model is:r(T)=(a+b×T)×[e(−(c+d×T))]
where *r*(*T*) is the developmental rate or daily fecundity, *T* is temperature or female adult age, depending on fitted data, *a*, *b*, *c*, and *d* are empirical constants.

### 2.4. Bieri-1 Model

This equation was used to fit the data of fecundity [37,41,45]. The Bieri–1 model uses the following equation:$$r\left(T\right)=\left[a\left(T-x_{max}\right)\right]-\left[b^{\left(T-x_{min}\right)}\right]$$
where *r*(*T*) is the developmental rate or fecundity, *T* is temperature or adult female age depending on the fitted data, *x_max_*, *x_min_*, *a*, and *b* are empirical constants (*x_max_*, *x_min_* represent the lower- and upper-temperature threshold or the days of beginning and ceasing oviposition).

### 2.5. Bieri-2 Model

This equation was used to fit the data on the developmental rate and fecundity by [7,41,45]. The Bieri–2 model uses the following equation:$$r\left(T\right)=a\times \left(\frac{\left(T-x_{min}\right)}{b^{\left(T-x_{min}\right)}}\right)$$
where *r*(*T*) is considered as developmental rate or daily fecundity, *T* is the temperature for developmental rate or female adult age for daily fecundity, depending on the fitted data, *x_min_*, *a*, and *b* are empirical constants.

### 2.6. Analytis Model

The Analytis model was first developed to determine the relationship between temperature and developmental time of phytopathogenic fungi and plant pests [46,47]. However, some authors used this model not only to describe development but also to determine the fecundity of insect pests and their natural enemies at different temperatures [40,41,44,45]. The expression of this model is:$$r\left(T\right)=a\times {\left(T-x_{min}\right)}^{n\;}\times {\left(x_{\max\;}-T\right)}^{m}$$
where *r*(*T*) is the developmental rate or fecundity, *T* is temperature or adult female age depending on the fitted data, and *x_max_*, *x_min_*, *a*, *m*, and *n* are empirical constants (*x_max_*, *x_min_*, represent the lower and the upper-thresholds for development or the first and the final age of oviposition).

### 2.7. Model Evaluation

All four non-linear models were assessed for goodness-of-fit to observed data [25] based on the following three statistical criteria:The coefficient of determination (*R*^2^), where RSS is the residual sum of squares and TSS is the total sum of squares. A higher value of *R*^2^ indicates a better fit.
$$R^{2}=1-RSS/TSS$$The residual sum of squares (*RSS*): A lower value of *RSS* indicates a better fit. The *R*^2^ is not an appropriate criterion to discriminate between models with different numbers of parameters (models with more parameters usually provide a better fit).The adjusted coefficient of determination (*R*^2^*_adj_*), where *n* is the number of observations, *p* is the number of model parameters, and *R*^2^ is the coefficient of determination. A higher value of *R*^2^*_adj_* indicates a better fit [44,45].
$$R_{adj}^{2}=1-\left[\frac{n-1}{n-p}\right]\left(1-R^{2}\right)$$

### 2.8. Statistical Analysis

The effect of temperature on the life history traits of *S. gilvifrons* [25] were analyzed with the age-stage, two-sex life table [48,49] using the computer program TWO SEXMSChart [50]. The parameters of the non-linear regressions for fecundity were analyzed with the Marquardt algorithm [51] using the JMP (IN 4.lnk) and Excel (version 2007) programs. All graphs were plotted with SigmaPlot 12.3.

## 3. Results and Discussion

This study provides fundamental information on the thermal performance in age-dependent fecundity of a widespread predator of herbivorous spider mites, that is, the ladybird *S. gilvifrons*. The models were fitted to oviposition and longevity data observed at six constant temperatures ranging from 15 to 34 °C [25].

The experimental study [25] underlying the modeling performed here suggested that oviposition of *S. gilvifrons*, as is the case in most ectothermic arthropods, is affected by temperature in a non-linear fashion, with the total number of eggs per female at 15, 20, 25, 27, 30, and 34 °C amounting to 20.5, 133.1, 159.6, 158.6, 249.5, and 221.2 eggs, respectively. These data were higher than those reported by [23] (25.5, 47.1, 102.5, 145.2, 150.9, and 185.5 eggs) at 15 to 35 °C, and differ considerably from those reported by [52], with averages of 30.5, 52.8 and 39.8 eggs at 20, 25, and 30 °C. Within the range of 15 to 35 °C, fecundity of *S. gilvifrons* is relatively high in comparison with other *Stethorus* spp., with *S. punctillum* being at the lower end (6.6 to 46.8 eggs at 16 to 32°C) [11] and *S. japonicas* being at the upper end (620.0 to 736.2 eggs at 20 to 30 °C [53]. Fecundities of the latter species represent by far the highest oviposition rates reported for any *Stethorus* spp.

In the experimental study [25] underlying the modeling presented here, the longevity of female *S. gilvifrons* ranged from 87.1 to 32.0 d at 15 to 34 °C [25], which is much longer than the longevities reported for this species by [23] at 15 to 35 °C (17.4 to 8.8 d), and by [52] at 20 to 30 °C (35.2 to 10.2 d). The reported differences in fecundities and longevities of the same species at similar temperatures may be due to differences in host plant and prey quality [28], photoperiod [54], humidity [55], or the techniques used in rearing and/or experiments.

Most non-linear models (Enkegaard, Bieri-1, Bieri-2, and Analytis) used to simulate fecundity of *S. gilvifrons* in our study showed significant goodness-of-fit according to *R*^2^, *R*^2^*_adj_*, and *RSS* values at most temperatures (Table 1; Figure 1). However, the fit of the models differed considerably among and within temperatures. All four models were better fitted near the optimum temperature (at about 30 °C) than at the other temperatures, which parallels modeling fecundity of two coccinellid *Nephus* spp. [41]. At 34 °C, all models had poor goodness-of-fit; the Bieri-1 and Analytis models had the highest *R*_2_ and *R*^2^*_adj_* values, whereas the Enkegaard model had the poorest fit to data (Table 1). The best fitting model varied with temperature: the Bieri-1 model had the best fit at 15 and 27 °C, whereas the Analytis model had the best fit at 20, 25, and 30 °C. 

The thermal performance models presented here can be used to simulate the daily, age-dependent egg production of *S. gilvifrons* under variable temperature conditions [56]. The high goodness-of-fit of the Bieri-1 and Analytis models at all temperatures except 34 °C indicates that these models adequately reflect the underlying biological processes of *S. gilvifrons*. Fluctuating temperatures may, but do not necessarily, result in different effects than constant temperatures in laboratory experiments, e.g., [57,58]; but see [59]. Due to Jensen’s inequality and typically non-linear thermal responses [35], such deviations occur primarily at temperatures close to and around the lower and upper thresholds but less so at the intermediate range [33,35]. Possible deviations also depend on the type of fluctuation. In fact, temperatures termed “constant” are usually also oscillating around the target mean by ±1, 2, or even more °C units, depending on the equipment and way of climate control (in our study it was the mean ±1 °C). Thus, in most laboratory settings “constant” temperatures are actually temperatures fluctuating at low amplitude around the mean. The amplitude and frequency of temperature changes, and smooth versus abrupt transition between temperatures are additional factors that may potentially affect the life history of the target organisms directly, by acting on its physiology and behavior, and/or indirectly via concurrent changes in diet, microclimate, humidity, etc. [35,57]. Notwithstanding accuracy trade-offs in predicting local/regional phenology, the review by [35] indicates that models based on close to constant temperatures are suitable and relevant for the planning and implementation of biological and integrated pest management programs. Thus, the fecundity models presented here can be useful for predicting the occurrence, population dynamics, phenology, and distribution patterns of *S. gilvifrons* under various environmental conditions and to estimate its establishment potential outside its native geographic range. Overall, our study suggests that, due to its thermal tolerance, *S. gilvifrons* can be considered a highly promising candidate for the biological control of spider mites such as *T. urticae* at a wide range of temperatures under field and greenhouse conditions [60,61,62].

## Figures and Tables

**Figure 1 insects-14-00199-f001:**
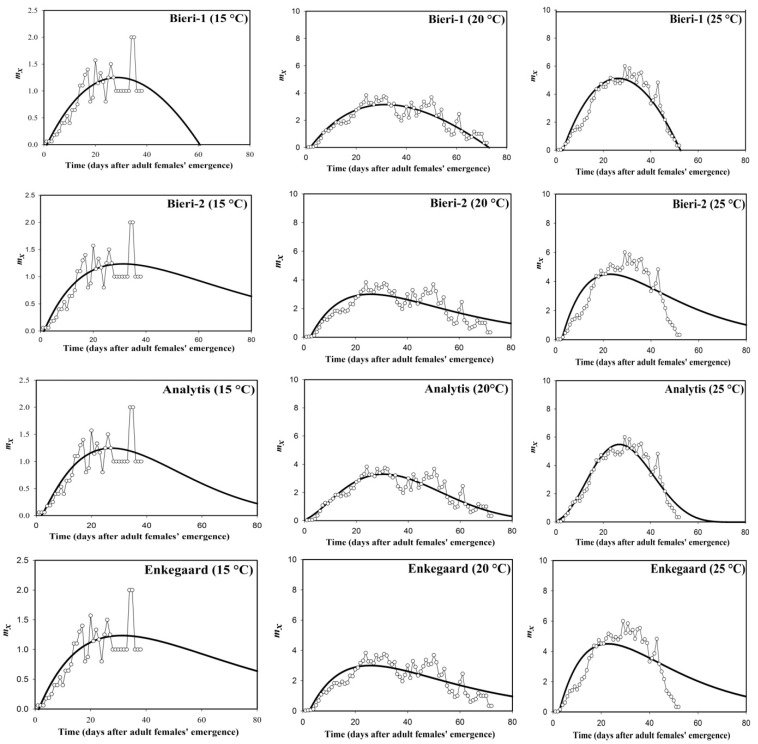
Fitting four non-linear models to observed age-specific fecundity (*m_x_*) of *S. gilvifrons* at six constant temperatures. Thin light lines and open circles represent the observed data [25].

**Table 1 insects-14-00199-t001:** Goodness-of-fit of four non-linear models to fecundity of the ladybird *S. gilvifrons* observed at six constant temperatures (observed data from [25]).

Model	N	Parameters	15 °C	20 °C	25 °C	27 °C	30 °C	34 °C
**Bieri-1**	4	*a*	−0.1824	−1.099	−0.1881	−1.2885	−0.418	−18.2343
		*b*	0.9844	0.9849	0.9726	0.9773	0.9094	0.9868
		*x* _max_	99.17	96.8981	84.5666	78.6850	44.4851	92.1040
		*x* _min_	184.95	309.2969	100.8353	221.4891	32.0370	562.1977
		*R* ^2^	0.6935	0.8804	0.8055	0.8180	0.8251	0.3507
		*R* ^2^ _adj_	0.6664	0.8729	0.7969	0.8059	0.8132	0.2482
		RSS(10^−4^)	2.84	21.80	16.65	68.80	78.56	1136.86
**Bieri-2**	3	*a*	0.1121	0.6027	0.3509	1.1872	1.6660	4.2338
		*b*	1.033	1.050	1.0437	1.0560	1.0890	1.1448
		*x* _min_	0.2814	2.4778	2.3966	1.6891	1.7523	5.8095
		*R* ^2^	0.6915	0.6878	0.7219	0.7525	0.7677	0.4052
		*R* ^2^ _adj_	0.6739	0.6656	0.7139	0.7418	0.7574	0.3457
		RSS(10^−4^)	2.87	163.53	25.47	102.32	123.33	5238.56
**Analytis**	5	*a*	3.17 × 10^−16^	1.47 × 10^−14^	1.72 × 10^−14^	8.53 × 10^−15^	2.18 × 10^−12^	8.86 × 10^−17^
		*D_min_*	1.7993	−4.1012	−1.9677	−0.3099	1.6813	1.8743
		*D_max_*	168.9471	81.0560	126.9143	111.8141	63.1247	66.1766
		*n*	6.4882	5.6469	5.7270	6.5432	6.1459	8.5172
		*m*	1.1649	3.2071	1.9327	1.6488	2.0061	2.2784
		*R* ^2^	0.6737	0.9152	0.8095	0.8014	0.9454	0.3349
		*R* ^2^ _adj_	0.6329	0.9080	0.7982	0.7833	0.9401	0.1784
		RSS(10^−4^)	2.7520	15.89	16.36	76.64	24.11	390.37
**Enkegaard**	4	*a*	−0.8739	−9.8291	−5.4350	−12.8136	−19.7534	−19.7533
		*b*	0.6820	3.9667	2.2659	7.5857	11.2724	12.7650
		*c*	1.7624	1.7624	1.7624	1.7624	1.7624	1.7624
		*d*	0.0333	0.0491	0.0428	0.0545	0.0853	0.0785
		*R* ^2^	0.6915	0.6786	0.7219	0.7525	0.7677	0.3921
		*R* ^2^ _adj_	0.6643	0.6586	0.7097	0.7360	0.7519	0.2961
		RSS(10^−4^)	2.87	63.53	25.47	102.32	123.33	333.64

N: Number of model parameters, *R*^2^: Coefficient of determination, *R*^2^*_adj_*: Adjusted *R*^2^, *RSS*: Residual sum of squares.

## Data Availability

The data presented in this study are available on request from the first author (M.J.).

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
