# Peer review of "Thermal Oviposition Performance of the Ladybird Stethorus gilvifrons Preying on Two-Spotted Spider Mites"

_insects, 2023, doi:10.3390/insects14020199_

Round 1
Reviewer 1 Report
This is a very short manuscript presenting the results of new statistical treatment of earlier published data on effects of constant temperatures on fecundity of a predatory ladybird Stethorus gilvifrons. The data were analyzed with 4 non-linear models and (depending on temperature) some models show somewhat better fit quality than the others. Experiments were well planned and conducted, statistical analysis is correct. The text is rather well written, I have only some minor corrections (see below).
However, the main problem of the paper is a very limited novelty: in fact, as was noted above, it is not an original study but new treatment of the results earlier published by the same authors. Besides, a plenty of papers comparing dozens of different models of various thermal effects have been published and I would not say that the present manuscript add something really new to these studies in the context of basic science. Indeed, the data on age-specific and total fecundity at different temperatures are important for biocontrol practitioners, but these data (as figures and tables) can be easily found in the earlier (original) publication by the same authors.
Lines 2-3: Latin names in the title also should be in italics font.
Line 70: Please, explain why you have selected namely these four from numerous models of thermal effects.
Lines 202-204: Please, increase the font size.
Author Response
This is a very short manuscript presenting the results of new statistical treatment of earlier published data on effects of constant temperatures on fecundity of a predatory ladybird Stethorus gilvifrons. The data were analyzed with 4 non-linear models and (depending on temperature) some models show somewhat better fit quality than the others. Experiments were well planned and conducted, statistical analysis is correct. The text is rather well written, I have only some minor corrections (see below).
However, the main problem of the paper is a very limited novelty: in fact, as was noted above, it is not an original study but new treatment of the results earlier published by the same authors. Besides, a plenty of papers comparing dozens of different models of various thermal effects have been published and I would not say that the present manuscript add something really new to these studies in the context of basic science. Indeed, the data on age-specific and total fecundity at different temperatures are important for biocontrol practitioners, but these data (as figures and tables) can be easily found in the earlier (original) publication by the same authors.
Due to conciseness, we changed the manuscript type from article to communication. This paper is novel and original because it is the first to model age- and temperature-dependent fecundity of the ladybird Stethorus gilvifrons and as such important to allow predicting the performance of this predator under various environmental conditions. Modeling processes such as age- and temperature-dependent fecundity are, in general, important efforts to detect and describe general patterns emerging from experimental data, which can then be used for prognosis. For these reasons, this paper is an important follow-up of the experimental paper by Jafari et al. (2020), which was published in Farsi. Splitting the modeling into the immature development phase and the oviposition phase is a mathematical requirement, no model can simultaneously cover both phases. Accordingly, also publication of these modeling efforts, particularly when different temperatures are compared, is commonly split in two papers. Presenting too many data and associated analyses and interpretations in one paper often comes at the expense of focus, conciseness and readability.
Lines 2-3: Latin names in the title also should be in italics font.
Corrected
Line 70: Please, explain why you have selected namely these four from numerous models of thermal effects.
Explanation added.
Lines 202-204: Please, increase the font size.
Done
Reviewer 2 Report
The authors have graphed and presented their results clearly, drawing some attention to the implications of their findings. The methods used are appropriate for the objectives of the work and, in general, well depicted. The resulting figures are sufficient, informative, and of good quality helping to follow the reasoning throughout the manuscript. The discussion of results and comments on future research should be improved if the paper is to be considered by Insects. The novelty of this work is also questionable. Major remarks have been made below for authors to consider.
My first concern is that the authors are extrapolating the applicability of their results beyond what the design supports. These are only development data from six sets of highly artificial constant temperatures ranging from 15C to 34C, so the inference power of the paper is very limited. This is a critical limitation of the study, and the authors must concede and discuss this, which they do indeed discuss, yet only briefly and not enough in my humble opinion.
It is well known that most of laboratory experiments are conducted under constant temperatures whereas in nature daily temperature fluctuations can be very wide. The interaction of cyclic temperatures with nonlinear function of development parameters can introduce significant deviations from the parameters obtained in this study. Studies at fluctuating temperatures are therefore encouraged so that more realistic effect of temperature on biological parameters of BCs could be understood, as this is the closest to temperature fluctuations that occur in the field. So, I am suggesting to the authors to tone-down the language a little and admit that there are still substantive uncertainties to be considered. Von Schmalensee and colleagues may disagree, but others may not, e.g., Journal of Economic Entomology 113:633-645. I'm missing a deeper discussion on this.
Some of the authors’ statements would be much stronger if they tie their work to the body of literature that has built up on the bio ecology of BCs parasitoids, e.g., Journal of Economic Entomology 112:1560-1574 and Journal of Economic Entomology 112:1062-1072. These studies, provide strong evidence that daily temperature fluctuations significantly affected development times (and longevity) of BCs, resulting in marked deviations and erroneous predictions when compared to their constant temperature counterparts. This article should provide details on all these fronts to provide the proper context for the work. This is not to diminish the data gathered in this study, as they are of value. But it is important for the authors not to overgeneralize, and to warn the reader, including regulatory agencies, against doing so as well. Adding these details will improve the discussion in my humble opinion.
My other concern is that the overall structure of this manuscript is very similar to that of their recently published paper (see reference #33 in the text). What really sets this paper apart from the other paper to deserve a separate publication in Insects? This is a classic case of 'microtome' or 'least publishable unit' manuscript writing. You should have included the information in this manuscript in that publication which would have made for a very nice single paper.
I hope you will consider revising with what I have noted in mind and resubmit.
Good luck!
Author Response
The authors have graphed and presented their results clearly, drawing some attention to the implications of their findings. The methods used are appropriate for the objectives of the work and, in general, well depicted. The resulting figures are sufficient, informative, and of good quality helping to follow the reasoning throughout the manuscript. The discussion of results and comments on future research should be improved if the paper is to be considered by Insects. The novelty of this work is also questionable. Major remarks have been made below for authors to consider.
My first concern is that the authors are extrapolating the applicability of their results beyond what the design supports. These are only development data from six sets of highly artificial constant temperatures ranging from 15C to 34C, so the inference power of the paper is very limited. This is a critical limitation of the study, and the authors must concede and discuss this, which they do indeed discuss, yet only briefly and not enough in my humble opinion.
It is well known that most of laboratory experiments are conducted under constant temperatures whereas in nature daily temperature fluctuations can be very wide. The interaction of cyclic temperatures with nonlinear function of development parameters can introduce significant deviations from the parameters obtained in this study. Studies at fluctuating temperatures are therefore encouraged so that more realistic effect of temperature on biological parameters of BCs could be understood, as this is the closest to temperature fluctuations that occur in the field. So, I am suggesting to the authors to tone-down the language a little and admit that there are still substantive uncertainties to be considered. Von Schmalensee and colleagues may disagree, but others may not, e.g., Journal of Economic Entomology 113:633-645. I'm missing a deeper discussion on this.
Some of the authors’ statements would be much stronger if they tie their work to the body of literature that has built up on the bio ecology of BCs parasitoids, e.g., Journal of Economic Entomology 112:1560-1574 and Journal of Economic Entomology 112:1062-1072. These studies, provide strong evidence that daily temperature fluctuations significantly affected development times (and longevity) of BCs, resulting in marked deviations and erroneous predictions when compared to their constant temperature counterparts. This article should provide details on all these fronts to provide the proper context for the work. This is not to diminish the data gathered in this study, as they are of value. But it is important for the authors not to overgeneralize, and to warn the reader, including regulatory agencies, against doing so as well. Adding these details will improve the discussion in my humble opinion.
We now elaborate on the issue of assessing life history traits under “constant” versus fluctuating temperature, and the relevance of such studies for prognosis of the performance in the field, in the discussion and added some pertinent references (see lines 204 to 224).
My other concern is that the overall structure of this manuscript is very similar to that of their recently published paper (see reference #33 in the text). What really sets this paper apart from the other paper to deserve a separate publication in Insects? This is a classic case of 'microtome' or 'least publishable unit' manuscript writing. You should have included the information in this manuscript in that publication which would have made for a very nice single paper.
Modeling immature development and oviposition must be mathematically split because no single model can simultaneously cover both phases. Accordingly, studies dealing with immature development models and those dealing with models of age-dependent fecundity pursue separate objectives and look for different traits and patterns. This alone warrants allocating these efforts to different papers, which is done quite commonly in this field. Also, presenting too many data and associated analyses and interpretations in one paper often comes at the expense of focus, conciseness and readability.
Round 2
Reviewer 2 Report
The manuscript presents a considerable amount of data worth being published. However, I’m still suggesting to the authors to tone-down the language a little and admit that there are still substantive uncertainties to be considered with respect to their methodology and reasoning.
As previously suggested, some of the authors’ statements would be much stronger if they tie their work to the body of literature that has built up on the bio ecology of BCAs reared under true fluctuating temperatures, e.g., Journal of Economic Entomology 112:1560-1574 and Journal of Economic Entomology 112:1062-1072. In those particular studies, each fluctuating temperature profile was modeled after field recorded temperatures that had the desired average target temperature. These are the first studies ever to undergo such analysis. In this regard, these studies, unlike the review by Von Schmalensee and colleagues, provide strong evidence that true daily temperature fluctuations significantly affected development times (and longevity) of BCAs, resulting in marked deviations and erroneous predictions when compared to their constant temperature counterparts. This article should provide details on all these fronts to provide the proper context for the work.
Again, this is not to diminish the data gathered in this study, as they are of value. But it is important for the authors not to overgeneralize, and to warn the reader, including regulatory agencies, against doing so as well. Adding these details will improve the discussion in my humble opinion.
The authors have further failed to elaborate on what sets this study apart from their previous findings/paper. I appreciate their honesty, and their response to my original comments will suffice, yet this also has to be explained in this article before the paper can be accepted for publication in Insects.
I hope the authors will revise with what I had in mind here and resubmit as the paper is quite good overall.
Best of luck to you!
Author Response
Comments and Suggestions for Authors
The manuscript presents a considerable amount of data worth being published. However, I’m still suggesting to the authors to tone-down the language a little and admit that there are still substantive uncertainties to be considered with respect to their methodology and reasoning.
As previously suggested, some of the authors’ statements would be much stronger if they tie their work to the body of literature that has built up on the bio ecology of BCAs reared under true fluctuating temperatures, e.g., Journal of Economic Entomology 112:1560-1574 and Journal of Economic Entomology 112:1062-1072. In those particular studies, each fluctuating temperature profile was modeled after field recorded temperatures that had the desired average target temperature. These are the first studies ever to undergo such analysis. In this regard, these studies, unlike the review by Von Schmalensee and colleagues, provide strong evidence that true daily temperature fluctuations significantly affected development times (and longevity) of BCAs, resulting in marked deviations and erroneous predictions when compared to their constant temperature counterparts. This article should provide details on all these fronts to provide the proper context for the work.
Again, this is not to diminish the data gathered in this study, as they are of value. But it is important for the authors not to overgeneralize, and to warn the reader, including regulatory agencies, against doing so as well. Adding these details will improve the discussion in my humble opinion.
We now explain our choice of temperature regime in the introduction (lines 67 to 77) and amended the pertinent section in the discussion (lines 213 to 231). This paper did not have the objective to compare the effects of mean temperatures generated by different methods but we discuss the advantages and disadvantages of using one or the other method. We appreciate the attempts by McCalla et al (2019) and Milosavljevic et al. (2019) to mimic natural conditions as closely as possible and refer to these studies in the introduction and discussion.
The authors have further failed to elaborate on what sets this study apart from their previous findings/paper. I appreciate their honesty, and their response to my original comments will suffice, yet this also has to be explained in this article before the paper can be accepted for publication in Insects.
The novel and original aspect of this study is the modeling part. As mentioned in the introduction (lines 78 to 89) and discussion (lines 178 to 181, 199 to 208), this is the first study to model temperature- and age-dependent fecundity of S. gilvifrons and to compare the goodness of fit of different models, in dependence of temperature As explained in the comments in the first reviewing round, the experimental paper (Jafari et al. 2020) reports the experimental data but does not contain any modeling. Furthermore, the experimental paper was published in Farsi (this is mentioned in the references) and is thus only accessible to Farsi-speaking colleagues; for this reason, we provide part of the experimental data in the discussion of this paper.